# The Impact of Thailand's Openness on Bilateral Trade between Thailand and Japan: Copula-Based Markov Switching Seemingly Unrelated Regression Model

**Pathairat Pastpipatkul [1,2], Petchaluck Boonyakunakorn [3,\*] and Kanyaphon Phetsakda [1]**

[1]  Faculty of Economics, Chiang Mai University, Chiang Mai 50200, Thailand; ppthairat@hotmail.com (P.P.); Kanyaphon_Phetsakda@cmu.ac.th (K.P.)
[2]  Center of Excellence in Econometrics, Faculty of Economics, Chiang Mai 50200, Thailand
[3]  Faculty of Agriculture, Chiang Mai University, Chiang Mai 50200, Thailand
\*  Correspondence: petchaluckecon@gmail.com

**Abstract:** The purpose of this paper is to analyze the impact of trade openness and the factors based on the gravity model on the bilateral trade flows between Thailand and Japan. The factors consist of GDP, distance, trade openness, and exchange rate. Bilateral trade is composed of two flows: Thailand's export flow to Japan, and Thailand's import flow from Japan. The specified gravity equations are estimated by Copula-based Markov switching seemingly unrelated regression approach. The best-fitting model is chosen based on the lowest Akaike information criterion (AIC) and Bayesian information criterion (BIC). The Normal and Student's t distributions are for Thailand's export equation and Thailand's import equation, respectively. The Student's t copula is applied for joint distribution. Analyzing the bilateral trade flow is separated into two situations, namely the high and the low growth markets. Empirical results show that distance provides a positive effect on the export in a high growth regime, but a negative impact on the export in a low growth regime. As for Thailand's import flow, all variables, but especially trade openness, provide strong evidence supporting significance for both regimes. For the GDPs of both Thailand and Japan, trade openness and the exchange rate increase import flow in a high growth market. Meanwhile, the exchange rate decreases import flow in a low growth market. The Markov Switching Probability Estimation notes that Thailand's trading with Japan is mostly in the fast-growing market.

**Keywords:** Thailand-Japan trade; gravity model; copula; Markov switching

**JEL Classification:** C13; F18; N75

## 1. Introduction

International trade in recent years has become one of the most important factors in developing Thailand's economy. Thailand's exports of goods and services in 2016 accounted for 68.40% of GDP and its imports accounted for 53.62% according to World Bank statistics (World Bank n.d.). From an economic perspective, international trade plays a crucial role in improving the economy of a country. It is stated that the export-led growth (ELG) hypothesis has been largely used in economic explanations by focusing on exports. This is because export expansion leads to a better allocation of scarce resources and generation of economies of scale, while significantly contributing to economic growth. Moreover, foreign currencies from trade provide a number of advantages. For example, more currencies can be converted into capital to invest in machineries with more advanced technology, leading to lower production costs. This leads to higher exports and therefore enhances economic growth (Bhagwati 1988).

Without doubt, trade depends on trade liberalization or trade openness. As it reduces trade restrictions and barriers, trade openness has been implemented by most of the developing and developed countries in the belief that it leads to a corresponding expansion in exports. The widely used trade barriers over the last few decades have consisted of quotas, import and export tariffs, and export subsidies. More availability for trade openness brings up a number of advantages to countries on both the import and export sides. Reducing trade barriers is the primary objective to being more competitive in the open market. Furthermore, trading countries have more opportunities to learn new advanced technologies from each other and make use of this knowledge in increasing productivity (Romer 1992; Barro and Sala-i-Martin 1995). Tang (2005) and Kepaptsoglou et al. (2010) provided empirical evidence of free trade agreements boosting trade among its trading members. Consequently, trade openness is likely the most important factor in increasing imports and exports. There are several previous studies related to trade liberalization enhancing economic growth and exports. For example, Joshi et al. (1996) and Semančíková (2016) have found that trade openness in developing countries has a positive impact on macroeconomic performance. Alesina et al. (2005) showed that both the size of trading countries and the size of the opening market share a substantial effect on economic growth. Thus, trade liberalization has been intensively implemented in Thailand for the past two decades. Nevertheless, studies carried out by Jenkins (1996) and Greenaway and Sapsford (1994) showed the negative relationship between trade liberalization and export performance.

Numerous empirical studies have analyzed the impacts of trade openness on growth, while only recently some works have concentrated on the effects of trade openness on trade flows. For example, the work of Gulzar (2016), which examined the impact of trade liberalization on the bilateral trade flows of ECO (Pakistan, Turkey and Iran) countries, showed that there was a long-running positive association between them. It also indicated that trade openness played a significant role in bilateral trade.

According to the report of The World Factbook (2019), Thailand was ranked as the third-largest exporter to Japan after China and the United States, which accounted for 9.5% of Thailand's total exports. As for Thailand's imports, Japan was the second-ranked import partner, lagging behind China, and accounting for 14.5% of total imports. The strong economic relationship between Thailand and Japan was the result of the Japan–Thailand Economic Partnership Agreement (JTEPA) established in November 2007 and the ASEAN-Japan Comprehensive Economic Partnership (AJCEP) signed in April 2008. The main advantage of such agreements is tariff reduction and granted special quotas, which bring a lot of benefits, such as increasing trade between countries. As a consequence, the JTEP agreement led to real GDP growth, better social welfare, and trade growth in Thailand. The principal merchandise exported to Japan are vehicles, equipment, processed chicken, faxes, and telephones. Meanwhile, Thailand mainly imports electrical and electronic equipment, machinery, nuclear reactors, boilers, iron, and steel.

Recent studies of international trade flows have shown that the gravity model has been recognized as a widely applicable approach (Kalirajan 2008). The model first introduced by Tinbergen in 1962 came from the gravity model of Newton. The gravity equation consists of the "mass" factor represented by the economic size, known as gross domestic product (GDP), and the distance represented by the geographical distance between two countries (Guo 2015). Some studies by Martínez-Zarzoso and Nowak in 2003, Simwaka in 2006, Achay in 2006, and Arabi and Ibrahim in 2012 have pointed out that GDP is found to be an important determinant of bilateral trade flows and has a positive impact on the volume of trade. For the distance variable, it directly refers to transaction costs consisting of shipping cost, the cost of obtaining information about outlaying economies, and the cost of contracting at a distance. As a consequence, a greater distance between partners leads to higher trading costs (Beugelsdijk et al. 2013). Hence, distance should have a negative relationship with trade, with supporting evidence from the results of studies carried out by Eichengreen et al. (1998) and Disdier and Head (2008). However, some works (Tinbergen 1962; Anderson and Wincoop 2003; Al-Majali and Adayleh 2018) have argued that distance is the friction of trade flows in the gravity model.

Moreover, international trade can be affected by many factors, such as trade-partner demand and supply, laissez-faire policy, the difference in technology, the distance between borders, diplomatic relationship, shared history and culture, and more. Some authors have extended the gravity model by considering other variables, such as exchange rate and infrastructure availability. For example, McKenzie (1998) investigated the impact of exchange-rate volatility on export growth. His empirical results showed that exchange-rate fluctuations could affect trade in either a positive or a negative direction. Baharumshah (2001) investigated the effect of the exchange rate on the bilateral trade balance of Malaysia and Thailand with the US and Japan and found that the real effective exchange rate (REER) is significant in the trade balance equation. In the long run, REER increases the trade balances of both trading partners. Some studies, however, reported no significance regarding the correlation between the exchange rate and export volume (Bahmani-Oskooee and Payesteh 1993; Hooper and Kohlhagen 1978). Senadza and Diaba (2017) have found that the exchange rate adversely affects exports in the short run, but found a positive relationship between them in the long run. Nevertheless, there is no significant impact of exchange rate volatility on imports.

The Copula-based Markov switching seemingly unrelated regression (MS-SUR) model is used to estimate bilateral trade between two countries with the gravity model. Conventionally, the errors of the SUR model are assumed to be normal distribution in all equations, which is a strong assumption. Boonyakunakorn et al. (2017) also allow the errors of the gravity trade model to have different distributions by using the copula model. Moreover, with the copula function, a multivariate distribution function can be decomposed into marginal distributions of each random variable and link each marginal distribution. The major advantage of copula is that marginal distribution functions can have different distributions. Finally, we extend SUR to be able to capture the dynamic change in the time series by using Markov switching (MS) since international trade mostly displays a different degree of dependence over time. In this study, we aim to focus further on the understanding of the effects of the trade openness on trade flow. We employ Japan as our case study since Japan is one of Thailand's most crucial economic partners in terms of both trade and investment. The paper consists of five sections. Section 2 outlines methodology. Section 3 discusses data. Section 4 presents unit root test and estimation, and the last section concludes.

## 2. Methodology

The factors in this study are GDP, distance, trade openness, and exchange rate. Bilateral trade flows include Thailand's export flows to Japan and Thailand's import flows from Japan.

### 2.1. The Gravity Model

The general form of linear gravity equation for international trade (Tinbergen 1962) is written by:

$$TRADE_{ij} = \beta_o + \beta_1 GDP_i + \beta_2 GDP_j + \beta_3 DIST_{ij} + \varepsilon_{ij} \tag{1}$$

where $TRADE_{ij}$ is bilateral goods trade flows that are exported and imported between origin $i$ and destination $j$. $\beta_o$ is the constant and does not depend on origin $i$ or destination $j$. $GDP_i$ is the natural log of the relevant gross domestic product of home country $i$. $GDP_j$ is the natural log of the gross domestic product of destination $j$. $DIST_{ij}$ is the distance between the capitals of the two countries.

The gravity equation specification of this work takes the form:

$$X_{TH,t} = \alpha_o + \alpha_1 GDP_{TH,t} + \alpha_2 GDP_{JP,t} + \alpha_3 DIST_{THJP,t} + \alpha_4 OPEN_{TH,t} + \alpha_5 EXC_{THJP,t} + \varepsilon_{1,t} \tag{2}$$

and

$$M_{TH,t} = \delta_o + \delta_1 GDP_{JP,t} + \delta_2 GDP_{JP,t} + \delta_3 DIST_{THJP,t} + \delta_4 OPEN_{JP,t} + \alpha_5 EXC_{THJP,t} + \varepsilon_{2,t} \tag{3}$$

where *TH* and *JP* denote Thailand and Japan, respectively, $t = 1, \ldots, T$, $\alpha_o$ and $\delta_o$ are constant terms, $\alpha(\alpha_1, \ldots, \alpha_5)$ and $\delta(\delta_1, \ldots, \delta_5)$ are coefficients, $\varepsilon_{1,t}$ and $\varepsilon_{2,t}$ are error terms, $X_{TH,t}$ represents the natural log of Thailand's export value to Japan at time $t$, and $M_{TH,t}$ represents the natural log of the value of Thailand's imports from Japan at time $t$.

The explanatory variables include:

- $GDP_{TH,t}$ and $GDP_{JP,t}$ are the growth of the gross domestic products of Thailand and Japan at time $t$, which represents the economic sizes of Thailand and Japan;
- $DIST_{THJP,t}$ is the gross domestic product weighted distance between Bangkok, Thailand and Tokyo, Japan. $DIST_{TH,t}$ is calculated by:

$$DIST_{TH,t} = \sum \left( dist_{TH,JP} \left( \frac{GDP_{TH,t}}{GDP_{TH,t} + GDP_{JP,t}} \right) \right) \tag{3a}$$

- $OPEN_{TH,t}$ and $OPEN_{JP,t}$ are the level of trade openness of Thailand and Japan at time $t$.

Trade openness in this paper is defined as the sum of exports and imports divided by GDP (Harrison 1996; Al-Majali and Adayleh 2018), which can be seen as the following:

$$OPEN_{TH,t} = \frac{total\ import_{TH,t} + total\ export_{TH,t}}{GDP_{TH,t}} \tag{3b}$$

and

$$OPEN_{JP,t} = \frac{total\ import_{JP,t} + total\ export_{JP,t}}{GDP_{JP,t}} \tag{3c}$$

Total trade openness is the indicator that better represents trade policies and trade relationships (Al-Majali and Adayleh 2018).

- $EXC_{THJP,t}$ is the exchange rate between the currency of Thailand (Thai Baht) and Japan (Japanese Yen) at time $t$.

## 2.2. Seemingly Unrelated Regression (SUR) Model

Zellner (1962) introduced the SUR model, which is useful in the study of a broad range of problems. The linear SUR model provides a more efficient parameter than the linear regression by involving a set of observations with cross-equation parameter restrictions and comparing correlated error terms with differing distributions (Zellner and Ando, 2010). The linear SUR equation is:

$$y_{i,t} = X_{i,t}\beta_{i,t} + \varepsilon_{i,t} \tag{4}$$

where $t = 1, \ldots, T$, $i = 1, \ldots m$, $y_{i,t}$ is the dependent variable, $X_{i,t}$ is the explanatory variables, $\beta_{i,t}$ is the coefficient, and $\varepsilon_{i,t}$ is the error term which denotes normal distribution with zero mean, variance $\sigma^2$, and variance–covariance matrix $\Omega$. If there are two equations, the linear SUR equations are:

$$y_{1,t} = X_{1,t}\beta_{1,t} + \varepsilon_{1,t} \tag{5}$$

and

$$y_{2,t} = X_{2,t}\beta_{2,t} + \varepsilon_{2,t} \tag{6}$$

The variance–covariance matrix is:

$$\Omega = \begin{bmatrix} \sigma_{11}I_T & \sigma_{21}I_T \\ \sigma_{12}I_T & \sigma_{22}I_T \end{bmatrix} = \Sigma \otimes I_T \tag{7}$$

where $\Sigma = \sigma_{ij}$, $i = 1$, and $j = 2$.

*2.3. Markov Switching Model with Seemingly Unrelated Regression (MS-SUR) Model*

From Equations (5) and (6), Markov switching with seemingly unrelated regression equation for the two equations can be written as:

$$y_{1,t} = \beta_{11}(S_t) + \beta_{12}(S_t)X_{1,1t} + \cdots + \beta_{1K}(S_t)X_{1,Kt} + \varepsilon_{1,t}(S_t) \tag{8}$$

and

$$y_{2,t} = \beta_{21}(S_t) + \beta_{22}(S_t)X_{2,1t} + \cdots + \beta_{2K}(S_t)X_{2,Kt} + \varepsilon_{2,t}(S_t) \tag{9}$$

where $S_t$ is the unobserved state variable with $K$ regimes which follow the first-order Markov process. $\varepsilon_{i,t}$ is the error term with 2 regimes. The structure of the error term is:

$$(S_t) = \begin{bmatrix} \varepsilon_{1,t}(S_t)\varepsilon_{1,t}(S_t) & \varepsilon_{1,t}(S_t)\varepsilon_{2,t}(S_t) \\ \varepsilon_{2,t}(S_t)\varepsilon_{1,t}(S_t) & \varepsilon_{2,t}(S_t)\varepsilon_{2,t}(S_t) \end{bmatrix} \tag{10}$$

The maximum likelihood approach is used to estimate Markov switching with seemingly unrelated regression functions. Suppose $S_t$ is known. Thus, the log-likelihood of Markov switching with seemingly unrelated regression for 2 regimes takes the form:

$$\ln L = \sum_{t=1}^{T}\left[ \sum_{j=1}^{K} \frac{MT}{2} \ln 2\pi - \frac{T}{2} \ln|\Sigma_t| \right]\left\{ -\frac{1}{2}\left[ Y_t - ((\beta_{MK}(S_t) + \beta_{MK}(1-(S_t)))X_t)' \right] \right.$$
$$\left. \Sigma_t'[Y_t - ((\beta_{MK}(S_t) + \beta_{MK}(1-(S_t)))X_t)] \right\} \Pr((S_t) = j) \tag{11}$$

where $M = 1,2$ which is the number of the equation. Suppose $S_t$ is unknown. Thus, the log-likelihood of Markov switching with seemingly unrelated regression for 2 regimes takes the form:

$$\ln L = \sum_{t=1}^{T}\left[ \ln \sum_{j=1}^{K} \left( f(y_t|(S_t) = j, \Theta)\Pr((S_t) = j) \right) \right] \tag{12}$$

where $\Theta = \{\beta_1(S_t), \beta_2(S_t), \sum(S_t)\}$ in each state, and the likelihood function is weighted by the probabilities of the state. $f(\cdot)$ is a density function corresponding to error assumption. However, the probabilities of the state are unknown. Assessing filter probabilities of each state $\Pr((S_t) = h)$ is given by Hamilton's filter.

Perlin (2015) provided the filter and the smoothing algorithm to calculate the filtered probabilities, which are based on the available data set. The steps for the forward filtering algorithm are as follows:

1.  Assume the initial value of transition probabilities $P_{ij}$. This probability refers to the switching probabilities from regime $i$ to regime $j$. The transition probabilities $P_{ij}$ can be written by:

$$P_{ij} = \Pr(S_{t+1} = j|S_t = i) \tag{13}$$

where $\sum_{j=1}^{h} P_{ij} = 1$ by $i, j = 1, \ldots, h$. The transition probabilities in the transition matrix **P** take the form:

$$\mathbf{P} = \begin{bmatrix} \begin{bmatrix} P_{11} & \cdots & P_{h1} \\ \vdots & \ddots & \vdots \\ P_{1h} & \cdots & P_{hh} \end{bmatrix} \end{bmatrix} \tag{14}$$

2.  Update the transition probability in order to compute the likelihood equation in each state $f(y_t|(S_t) = j, \varphi_{t-1})$ based on not only previous information, but also on all the parameters which

are in the equation consisting of $\Theta_{t-1}$ and $P_{ij}$. The form of the updated probability for each state is:

$$\Pr(S_t = j|\varphi) = \frac{f(y_t|S_{t-j}, \varphi_{t-1})\Pr(S_{t-j}|\varphi_{t-1})}{\sum_{j=1}^{k} f(y_t|S_{t-j}, \varphi_{t-1})\Pr(S_{t-j}|\varphi_{t-1})} \tag{15}$$

where $\varphi_t$ is the available information at time $t$ in the matrix form.

3. Iterate both step 1 and step 2 for $t = 1, \ldots, T$

### 2.4. Copula Model

The Copula model is widely used to explain the dependence structure between the variables. Therefore, to allow the marginal distribution functions to have different forms (Sklar 1973), we apply the copula to a joint distribution of random variables. Joint distribution with d-dimension ($H$) of a random variable ($X_n$) is derived by decomposing into marginal distribution $F_n$ and d-copula $C$, as follows:

$$H(X_1, \ldots, X_d) = C(F_1(X_1), \ldots, F_d(X_d)) \tag{16}$$

Equation (16) is the multivariate distribution function. The marginal distributions are continuous. In general, the models of the marginal distribution and the joint dependence have separate forms. Thus, copula function takes the form:

$$C(u_1, \ldots, u_d) = C\left(F_1^{-1}(u_1), \ldots, F_d^{-1}(u_d)\right) \tag{17}$$

where $u$ is the uniform distribution with range [0,1]. The D-dimension derives form a density copula as follows:

$$h(x_1, \ldots, x_d) = C(F_1(x_1), \ldots, F_d(x_d)) \tag{18}$$

The copula families that are used in this paper are Elliptical and Archimedean. The dependency of symmetry on both the left and right tails is captured by the Elliptical copula family. The Archimedean copula family can capture only one tail dependence. The Elliptical copula family includes the "Normal copula and the Student's t copula". The Archimedean copula family includes "Gumbel, Clayton, Joe, and Frank". Hence, this paper uses five copulas, which are two elliptical families (Normal copula and Student's t copula), and three Archimedean families (Clayton, Gumbel, and Joe).

### 2.5. Copula-Based Markov Switching Seemingly Unrelated Regression (MS-SUR) Model

The Copula-based Markov switching seemingly unrelated regression model allows the residuals of any equations to have different distributions (Pastpipatkul et al. 2016; Boonyakunakorn et al. 2017). The gravity of Thailand's export equation (2) and the gravity of Thailand's import equation (3) with 2 regimes can be written as:

$$X_{TH,t}(S_t = 1) = \alpha_o + \alpha_1 GDP_{TH,t} + \alpha_2 GDP_{JP,t} + \alpha_3 DIST_{THJP,t} + \alpha_4 OPEN_{TH,t}$$
$$+ \alpha_5 EXC_{THJP,t} + \varepsilon_{1,1t} \tag{19}$$

$$X_{TH,t}(S_t = 2) = \beta_o + \beta_1 GDP_{TH,t} + \beta_2 GDP_{JP,t} + \beta_3 DIST_{THJP,t} + \beta_4 OPEN_{TH,t}$$
$$+ \beta_5 EXC_{THJP,t} + \varepsilon_{1,2t} \tag{20}$$

$$M_{TH,t}(S_t = 1) = \delta_o + \delta_1 GDP_{JP,t} + \delta_2 GDP_{TH,t} + \delta_3 DIST_{THJP,t} + \delta_4 OPEN_{JP,t}$$
$$+ \delta_5 EXC_{THJP,t} + \varepsilon_{2,1t} \tag{21}$$

$$M_{TH,t}(S_t = 2) = \gamma_o + \gamma_1 GDP_{JP,t} + \gamma_2 GDP_{TH,t} + \gamma_3 DIST_{THJP,t} + \gamma_4 OPEN_{JP,t}$$
$$+ \gamma_5 EXC_{THJP,t} + \varepsilon_{2,2t} \tag{22}$$

Using the chain rule to construct the copula-based MS-SUR likelihood:

$$\frac{\partial^2}{\partial u_1(S_t)\partial u_2(S_t)}F(u_1(S_t)\partial u_2(S_t)) = f_1(u_1(S_t))f_2(u_2(S_t))c(u_1(S_t), u_2(S_t)) \tag{23}$$

Assume that $u_1(S_t)$ and $u_2(S_t)$ are the marginal distributions, which are either Normal distribution nor Student-t distribution for $S_t = 1, 2$. It is transformed into uniform distribution (0, 1) from Thailand's export and Thailand's import variables. $f_1(u_1(S_t))$ and $f_2(u_2(S_t))$ are the margins of Thailand's export and import variables, while the density function is $c(u_1(S_t), u_2(S_t))$ of Gaussian, Student-t, Clayton, Gumbel, and Joe for the link between Thailand's export and import equations. The logarithm is used to transform Equation (23). The likelihood in Equation (23) is multiplied with the likelihood in Equation (14) in order to have the full likelihood function of the MR-SUR model-based copula. Thus, the log-likelihood function is:

$$\ln L = \sum_{t=1}^{T}\left[\ln\sum_{j=1}^{2}\left(f(y_t|(S_t) = j, \Theta)\Pr((S_t) = j|\varphi_t)\right)\right] \tag{24}$$

where $f(y_t|(S_t) = j, \Theta)$ is the joint density between Equation (23) and $\Pr((S_t) = j|\varphi_t)$, which is defined as the probability of filter which comes from Equation (13), and $\Theta$ is defined as the parameters from the copula-based MS-SUR model.

## 3. Data

The datasets of Thailand and Japan from 1989 to 2017 are on a quarterly basis. Thailand's exports to Japan and Thailand's imports from Japan were collected from the World Integrated Trade Solution (WITS n.d.) database. The explanatory variables are GDP, distance, level of trade openness, and exchange rate. The GDP, total import, and total export were obtained from the World Bank database. Distance between Thailand and Japan was measured using Google Maps. Finally, the exchange rate was obtained from the Bank of Thailand (BOT).

## 4. Empirical results

*Stationary Process*

An augmented Dickey–Fuller (ADF) test and Phillips–Perron (PP) test were used to test the unit root in the stationary process. The null hypothesis of the ADF and PP tests is that the time-series is stationary and has a unit root. Table 1 shows the unit root test results of the ADF and PP test statistics at the level and 1st difference. Wasserstein and Lazar (2016) from the American Statistical Association (ASA) demonstrated that *p*-value does not provide a good measurement in statistic testing. As a consequence, this study presents the Bayes factor as an alternative approach for testing accurate null hypotheses. Overall, the Bayes factor provides some evidence to support that all variables are stationary at the first difference.

Table 2 shows the selecting of the best model for MS-SUR model from the lowest Akaike information criterion (AIC) and Bayesian information criterion (BIC). Hence, the best-fitting model is model 6, obtaining Normal and Student-t distributions for Thailand's export and import equations, respectively, and the student-t copula for joint distribution.

**Table 1.** Unit root test result. ADF test: augmented Dickey–Fuller test; PP test: Phillips–Perron test.

| Variable | ADF Test | | | | PP Test | | | |
|---|---|---|---|---|---|---|---|---|
| | Level | | 1st Diff | | Level | | 1st Diff | |
| | Intercept | Trend and Intercept | Intercept | Trend and Intercept | Intercept | Trend and Intercept | Intercept | Trend and Intercept |
| EXP | −2.418 (0.065) [0.483] | −2.886 (0.284) [0.972] | −10.251 *** (0.000) [0.000] | −8.859 *** (0.000) [0.000] | −2.879 (0.051) [0.414] | −2.886 (0.171) [0.822] | −10.429 *** (0.000) [0.000] | −11.526 *** (0.000) [0.000] |
| IMP | −1.367 (0.595) [0.839] | −2.019 (0.584) [0.854] | −8.378 *** (0.000) [0.000] | −8.337 *** (0.000) [0.000] | −1.482 (0.051) [0.906] | −2.342 (0.408) [0.994] | −8.290 *** (0.000) [0.000] | −8.246 *** (0.000) [0.000] |
| GDP$_{TH}$ | −0.763 (0.825) [0.432] | −2.282 (0.440) [0.982] | −9.958 *** (0.000) [0.000] | −9.924 *** (0.000) [0.000] | −0.763 (0.825) [0.431] | −2.505 (0.325) [0.993] | −9.967 *** (0.000) [0.000] | −9.935 *** (0.000) [0.000] |
| GDP$_{JP}$ | −0.977 (0.759) [0.568] | −2.543 (0.307) [0.986] | −8.962 *** (0.000) [0.000] | −8.931 *** (0.000) [0.000] | −1.009 (0.748) [0.591] | −2.711 (0.235) [0.925] | −8.909 *** (0.000) [0.000] | −8.875 *** (0.000) [0.000] |
| DIST | −1.533 (0.513) [0.931] | −2.259 (0.513) [0.931] | −7.845 *** (0.000) [0.000] | −7.815 *** (0.000) [0.000] | −1.260 (0.646) [0.768] | −1.968 (0.612) [0.818] | −7.884 *** (0.000) [0.000] | −7.758 *** (0.000) [0.000] |
| OPEN$_{TH}$ | −1.847 (0.356) (0.999) | −2.480 (0.337) [0.996] | −8.492 *** (0.000) [0.000] | -8.506 *** (0.000) [0.000] | −0.763 (0.193) [0.863] | −2.505 (0.244) [0.935] | −9.967 *** (0.000) [0.000] | −9.935 *** (0.000) [0.000] |
| OPEN$_{JP}$ | −2.251 (0.190) [0.858] | −2.586 (0.288) [0.974] | −8.684 *** (0.000) [0.000] | −8.725 *** (0.000) [0.000] | −1.845 (0.357) [1.000] | −2.614 (0.275) [0.965] | −8.884 *** (0.000) [0.000] | −8.898 *** (0.000) [0.000] |
| EXC | −2.778 (0.065) [0.483] | −2.592 (0.285) [0.972] | −7.825 *** (0.000) [0.000] | −7.935 *** (0.000) [0.000] | −2.830 (0.058) [0.447] | −2.314 (0.423) [0.990] | −7.877 *** (0.000) [0.000] | −7.830 *** (0.000) [0.000] |

Note that *** denotes the Augmented Dickey–Fuller test statistic and the Phillips–Perron test statistic significant at 1 level %. *p*-value is in the (.) and [.] is the Bayes factor computed by $-\exp(1)p\log p$ for $p < 1/\exp(1)$ (Sellke et al. 2001). The Bayes factor contributes to supporting the level of evidence in which BF < 1 into levels of evidence against. Strength of evidence against includes weak (1–0.333), moderate (0.333–0.1), substantial (0.1–0.033), strong (0.033–0.01), very strong (0.01–0.003), and decisive (<0.003) (Held and Ott 2016).

**Table 2.** Information criterion for model selection.

| Model | 1 | 2 | Copula | AIC | BIC |
|:-----:|:--:|:--:|:------:|:----:|:----:|
| 1 | Normal | Normal | Gaussian | −1431.922 | −1347.611 |
| 2 | Normal | Student-t | Gaussian | −1353.038 | −1263.457 |
| 3 | Student-t | Normal | Gaussian | −1331.857 | −1242.276 |
| 4 | Student-t | Student-t | Gaussian | −1441.685 | −1346.835 |
| 5 | Normal | Normal | Student-t | −1282.491 | −1198.180 |
| 6 | Normal | Student-t | Student-t | −1484.628 * | −1389.778 |
| 7 | Student-t | Normal | Student-t | −1357.210 | −1267.630 |
| 8 | Student-t | Student-t | Student-t | −1352.715 | −1263.134 |
| 9 | Normal | Normal | Clayton | −1089.794 | −1005.483 |
| 10 | Normal | Student-t | Clayton | −934.110 | −844.529 |
| 11 | Student-t | Normal | Clayton | −935.906 | −846.325 |
| 12 | Student-t | Student-t | Clayton | −1119.558 | −1024.708 |
| 13 | Normal | Normal | Gumbel | −920.353 | −836.042 |
| 14 | Normal | Student-t | Gumbel | −917.966 | −828.385 |
| 15 | Student-t | Normal | Gumbel | −916.802 | −827.222 |
| 16 | Student-t | Student-t | Gumbel | −915.163 | −820.313 |
| 17 | Normal | Normal | Joe | −939.800 | −855.489 |
| 18 | Normal | Student-t | Joe | −936.221 | −846.641 |
| 19 | Student-t | Normal | Joe | −937.162 | −847.581 |
| 20 | Student-t | Student-t | Joe | −934.500 | −839.650 |

Note that * indicates the lowest AIC and BIC.

Table 3 shows the estimated results of Thailand's exports to Japan in regime 1, which is defined as a high growth market, and regime 2, which is defined as a low growth market. For Thailand's export equation in the high growth market (regime 1), the Thai GDP and Japanese GDP positively affect Thai exports to Japan. The result was according to expectation. The coefficients of the Thai GDP and Japanese GDP are 0.512 and 0.470, respectively. This means that a 1% increase in the Thai GDP and Japanese GDP would lead to an increase in Thailand's exports of 0.512% and 0.470%, respectively. In a low growth market, the Thai GDP and Japanese GDP also positively affect Thai exports to Japan. This means that should the Thai GDP and the Japanese GDP increase by 1%, Thai exports to Japan will increase at 0.527% of Thailand's GDP and 0.029% of Japan's GDP. This implies that in a high growth regime, the Thai GDP has more influence on Thai exports than in a low one. Meanwhile, the effect of Japan's GDP on exports in a high growth regime is much lower than in a low growth regime. Thus, this indicates that the amount of Thailand's exports to Japan mainly depends on Japan's economic condition. As Japan's GDP increases, the demand for goods increases, which in turn leads to a rise in the export demand. This result is compatible with the works of Tinbergen (1962) and Anderson and Wincoop (2003).

**Table 3.** The results of Thailand's export equation.

| Variables | Regime 1 | | | Regime 2 | | |
|:---------:|:-----------:|:--------:|:--------:|:-----------:|:--------:|:--------:|
| | Coefficient | Std. Err | B Factor | Coefficient | Std. Err | B Factor |
| Intercept | 0.002 | 0.012 | 0.317 | 0.002 | 0.014 | 0.310 |
| GDPTH | 0.512 | 0.838 | 0.903 | 0.527 | 2.615 | 0.397 |
| GDPJP | 0.470 | 1.699 | 0.522 | 0.029 | 0.385 | 0.156 |
| DIST | 0.003 | 0.282 | 0.025 | −0.007 | 0.271 | 0.057 |
| OPNSTH | 2.233 | 2.645 | 0.997 | 3.649 | 2.696 | 0.831 |
| EXC | 0.342 | 0.607 | 0.868 | −0.324 | 0.646 | 0.812 |
| Sigma | 0.023 | 0.004 | 0.000 | 0.067 | 0.073 | 1.000 |

Note that B factor is Bayes factor which BF < 1 into levels of evidence against $H_0$. The strength of evidence against $H_0$ includes weak (1–0.333), moderate (0.333–0.1), substantial (0.1–0.033), strong (0.033–0.01), very strong (0.01–0.003), and decisive (<0.003) (Held and Ott 2016).

The distance variable shows a positive and significant estimate in a high growth market. The coefficient for distance is 0.003, thus a 1% increase in distance causes a 0.003% increase in Thailand's exports. On the other hand, a low growth market shows that distance negatively affects exports, with a −0.007 coefficient.

The results from both regimes provide a small value for the coefficients. Hence, it can be concluded that the distance variable has little effect on Thailand's exports, in line with the work of Hammer Strømman and Duchin (2006). They reported that the cost of distance has little impact on a region's total imports or exports of a given commodity. In a fast-growing market, a country's exports slightly concern the distance. Thailand still exports more even though the cost of distance increases, while a slow-growing market provides a small decrease in Thailand's exports.

The results for trade openness provide a positive impact on Thailand's exports in both high and low growth markets. The coefficients of trade openness are 2.233 and 3.649 for Thailand and Japan respectively. It can be seen that trade openness provides the benefits of trade, especially in a slow-growth market. Gulzar (2016) demonstrated that trade openness plays a crucial role in bilateral trade. For the exchange rate, the result was found positive and significant in a high growth market, with a coefficient of 0.342. A 1% increase in the exchange rate would correspond to a 0.342% rise of Thailand's exports. By contrast, the result shows a negative effect and significance, with a coefficient of −0.324 in a low growth market.

Table 4 shows the estimated result of Thailand's import equation in regime 1 referring to a high growth market, with regime 2 representing the low growth market. For Thailand's import equation in a high growth market, the Japanese GDP and Thai GDP positively affect Thai imports from Japan. The coefficient for the Japanese GDP and Thai GDP are 0.409 and 0.231, respectively. In a low growth market, the Thai GDP and Japanese GDP also positively affect Thai imports from Japan, with coefficients of 0.259 and 0.149, respectively. Since imports always depend on income level, as an increase in domestic income level accelerates the demand for imports (Aqeel and Butt 2001), the empirical finding seems to go in the direction of the recent work of Martínez-Zarzoso and Martínez-Zarzoso and Nowak-Lehmann (2003); Simwaka (2006); Achay (2006); and Arabi and Ibrahim (2012), which found a positive impact of GDP on the volume of trade.

**Table 4.** The results of Thailand's import equation.

| Variables | Regime 1 | | | Regime 2 | | |
|---|---|---|---|---|---|---|
| | Coefficient | Std. Err | B Factor | Coefficient | Std. Err | B Factor |
| Intercept | 0.055 | 1.161 | 0.001 | 0.282 | 2.132 | 0.029 |
| GDPJP | 0.409 | 2.739 | 0.000 | 0.259 | 2.463 | 0.000 |
| GDPTH | 0.231 | 2.535 | 0.000 | 0.140 | 1.274 | 0.000 |
| DIST | −0.029 | 3.117 | 0.000 | 0.100 | 8.673 | 0.002 |
| OPNSJP | 1.048 | 4.329 | 0.000 | 1.808 | 1.274 | 0.003 |
| EXC | 0.191 | 1.647 | 0.000 | −0.183 | 3.246 | 0.001 |
| SIGMA | 0.557 | 3.111 | 0.039 | 0.557 | 1.717 | 0.069 |

Note that B factor is Bayes factor which BF < 1 into levels of evidence against $H_0$. Strength of evidence against $H_0$ includes weak (1–0.333), moderate (0.333–0.1), substantial (0.1–0.033), strong (0.033–0.01), very strong (0.01–0.003), and decisive (<0.003) (Held and Ott 2016).

The distance variable shows a positive and significant estimate in a high growth market, where the coefficient for distance is −0.029. On the other hand, in a low growth market, the distance variable positively affects imports, with a coefficient of 0.100. The result of trade openness provides strong and significant positive effects on Thailand's imports in both high and low growth markets. The coefficients for trade openness are 1.048 and 1.808. These results confirm the advantage of the Thailand–Japan free trade agreement for Thailand's imports from Japan. The result for the exchange rate shows positive and significant attributes in a high growth market, with a coefficient of 0.191. However, the result shows a negative effect and significance, with a coefficient of −0.183 in a low growth market.

Table 5 shows the probability of being in regime 1 and regime 2, which is 0.999 and 0.599, respectively. We can conclude that trade between Thailand and Japan takes place mainly in a high growth economy. As can be seen, in the period of this study Thailand's trade with Japan ranked as the second-largest exporter and the third-largest importer in 2018.

**Table 5.** Markov switching probability estimation.

| Switching Probability | Coefficient | Std. Err | B Factor |
|:---:|:---:|:---:|:---:|
| $P_{11}$ | 0.999 | 2.201 | 0.09 |
| $P_{22}$ | 0.559 | 9.895 | 0.12 |

Note that B factor is Bayes factor which BF < 1 into levels of evidence against.

## 5. Conclusions

This empirical study examines the impact of trade openness and the factors based on the gravity model on bilateral trade flows between Thailand and Japan from 1989 through 2017 by using the Copula-based Markov switching seemingly unrelated regression approach. The best-fitting model is chosen by the lowest AIC and BIC, which provides the Normal and Student-t distribution for Thailand's export equation and Thailand's import equation, respectively. The Student's t copula is applied for joint distribution. There are two regimes applied in this study, namely a high growth market and a low growth market.

From the estimation of Thailand's exports in the high growth market, all variables provide a positive relationship with Thailand's exports to Japan. In a low growth market, Thailand's exports are negatively affected by distance and exchange rates. However, only distance offers strong evidence in support of significance for both regimes. For the import equation in a high growth market, all variables except for distance have a positive relationship with Thailand's imports. In a low growth market, only the exchange rate presents negative effects on imports. On the other hand, other variables provide a positive influence. Furthermore, the results for both high and low regimes provide strongly supporting significance, especially the trade openness variable. The Markov switching probability estimation notes that Thailand's trading with Japan mostly takes place in the fast-growing regime. The result suggests that the Thai government should support trade openness and promote economic growth, which enhances trade flows in a high growth market. Additionally, the government should encourage the ability of trading in the slow-growing market as well.

**Author Contributions:** Conceptualization, P.P.; Data curation, K.P.; Methodology, P.P. and P.B.; Writing—original draft, K.P.; Writing—review & editing, P.B. All authors have read and agreed to the published version of the manuscript.

**Funding:** This research received no external funding.

**Conflicts of Interest:** The authors declare no conflict of interest.

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
