# Peer review of "The Impact of Thailand’s Openness on Bilateral Trade between Thailand and Japan: Copula-Based Markov Switching Seemingly Unrelated Regression Model"

_economies, doi:10.3390/economies8010009_

Round 1

Reviewer 1 Report

The paper analyses the impact of trade openness, and the factors based on the gravity model on the bilateral trade flows between Thailand and Japan. The authors specify the gravity equations using Copula-based Markov switching seemingly unrelated regression approach. Whiles their adopted method and analysis are profound, I found the following comments to be attended to.

The introduction does not make a strong case for the manuscript. It lacks any theoretical foundation as a basis for the paper and proper literature works. The first paragraph for instance talks about Thailand and Japan trade relationship without theoretical basis and proper citation. “The world integrated trade solution reported that Japan was the third rank of the most imported products from Thailand after the United States and China which account 9.39 percent share of Thailand’s total products exports in 2015.” where do we find this? Paragraph two throw a little light on trade openness by first defining it. However, the definition given is not enough to prove theoretical backing given the fact the paper analyses the impact of trade openness. The authors should elaborate on this impact also theoretically, rather than just stating literature on what trade does. What are the scope and privileges in terms of concessions that are granted to each country. e.g. trade barriers. It will useful for readers to know the state of trade openness and its impacts for the two countries. I suggest for the authors to include, possibly a section that provides a general overview of trade openness and its trend impacts, the bilateral trade, its structure and some literature for the two countries. It seems the objectives of the paper are not clearly stated and as a result, the dynamics of trade openness and its impact fall short in the paper. E.g. How does trade Openness affect growth (GDP) from the viewpoint of foreign direct investment and domestic market among these two countries?. The Bhagwati (1995) growth led hypothesis underscore  different impact of trade between developed and less developed countries. Can the authors explain how this affect GDP in between Thailand and Japan?. The following references could be useful and worthy of citation: Alesina, A., Spolaore, E., & Wacziarg, R. (2005). Trade, growth and the size of countries. In Handbook of economic growth (Vol. 1, pp. 1499-1542). Asamoah, L. A., Mensah, E. K., & Bondzie, E. A. (2019). Trade openness, FDI and economic growth in sub-Saharan Africa: do institutions matter?. Transnational Corporations Review11(1), 65-79. Thurbon, E., & Weiss, L. (2006). Investing in openness: The evolution of FDI strategy in South Korea and Taiwan. New Political Economy11(1), 1-22. The paper needs serious English editing and polishing. Many sentences should be rewritten and spelling mistakes should be checked. e.g.  fourth not forth. Again The title itself is not clear: “Analysing Bilateral Trade Thailand with Japan…”. P8, L251; “Analyzing the impact of trade openness, and the factors based on the gravity model on bilateral trade flows between Thailand and Japan by using Copula-based Markov switching seemingly unrelated regression approach.” This is not a complete sentence and has not meaning. P7, L22; Thailand’s export represents that Thailand exports products to Japan. ??? P7, L255;  This result consists of the work of Tinbergen, 1962; …. What is the meaning. Also, starting sentences with “After that”, “Then” should be avoided P4, L14; Remove external square bracket P6, L198, TFP is not defined P1 and others!! What is Thai export, Thai GDP? Find difference between Thai and Thailand!! The results need to be more clarified: The authors should result for the ADF and PP to see how the variables became stationary after first difference. Does regime 1 corresponds to high growth market? Add respectively to end the sentence on P7, L222. Tables 2 and 3: The variables do not correspond to the variables defined for the equations. I suppose The TH and JP as used in the Tables must be subscript. Please check and correct. Also, “For the Thailand’s export equation in the high growth market, the Thai GDP and the level of Thailand’s trade openness affect positively Thai exports”  How? It is not clear what indicators support this assertion. Authors should explain statement with results from the table. The authors should do the following: 1) Elaborate more on the results from the table with economic intuition. Stating regression results only and citing literature works to back result is not enough. 2) Authors should comment on the implication of the coefficient of other variables. Example, What does the negative correlation of DIST and TFP to export in Regime implies? 2). Results in Regime are completely left out in the paper. Can the authors explain what is happening in Regime 2 for readers to know? P8, L247; “Table 4 shows Markov Switching probability in regime 1 and regime 2. Coefficients of Markov Switching Probability in regime 1 and regime 2 are very similar which are 0.323 and 0.323, respectively.” How does this result statement add to contribution in the paper? Regimes 1 and 2 have similar Coefficients of Markov Switching Probability means what? The phrase “boost decisively” as used in the abstract and conclusion is misleading. It tends to imply causality which is not backed by the result. The authors should write as obtained from the results … “positive correlation”.

Author Response

Response to Reviewer 1 Comments

Point 1: The introduction does not make a strong case for the manuscript. It lacks any theoretical foundation as a basis for the paper and proper literature works. The first paragraph for instance talks about Thailand and Japan trade relationship without theoretical basis and proper citation. “The world integrated trade solution reported that Japan was the third rank of the most imported products from Thailand after the United States and China which account 9.39 percent share of Thailand’s total products exports in 2015.” where do we find this?

Response 1: We have added the theoretical background in introduction part.

“The world integrated trade solution reported that Japan was the third rank of the most imported products from Thailand after the United States and China which account 9.39 percent share of Thailand’s total products exports in 2015.” We find it from CIA which we have already added in our paper.

Point 2: two throw a little light on trade openness by first defining it. However, the definition given is not enough to prove theoretical backing given the fact the paper analyses the impact of trade openness. The authors should elaborate on this impact also theoretically, rather than just stating literature on what trade does. What are the scope and privileges in terms of concessions that are granted to each country, e.g. trade barriers. It will useful for readers to know the state of trade openness and its impacts for the two countries. I suggest for the authors to include, possibly a section that provides a general overview of trade openness and its trend impacts, the bilateral trade, its structure and some literature for the two countries.

Response 2: We have already added a general overview of trade openness and some literature for the two countries.

Point 3: It seems the objectives of the paper are not clearly stated and as a result, the dynamics of trade openness and its impact fall short in the paper. E.g. how does trade Openness affect growth (GDP) from the viewpoint of foreign direct investment and domestic market among these two countries? The Bhagwati (1995) growth led hypothesis underscore different impact of trade between developed and less developed countries. Can the authors explain how this affect GDP in between Thailand and Japan? The following references could be useful and worthy of citation: Alesina, A., Spolaore, E., & Wacziarg, R. (2005). Trade, growth and the size of countries. In Handbook of economic growth (Vol. 1, pp. 1499-1542). Asamoah, L. A., Mensah, E. K., & Bondzie, E. A. (2019). Trade openness, FDI and economic growth in sub-Saharan Africa: do institutions matter? Transnational Corporations Review, 11(1), 65-79. Thurbon, E., & Weiss, L. (2006). Investing in openness: The evolution of FDI strategy in South Korea and Taiwan. New Political Economy, 11(1), 1-22.

Response 3: We have added more literature review.

Point 4 The paper needs serious English editing and polishing. Many sentences should be rewritten and spelling mistakes should be checked. e.g.  fourth not forth. Again The title itself is not clear: “Analysing Bilateral Trade Thailand with Japan…”. P8, L251; “Analyzing the impact of trade openness, and the factors based on the gravity model on bilateral trade flows between Thailand and Japan by using Copula-based Markov switching seemingly unrelated regression approach.” This is not a complete sentence and has not meaning. P7, L22; Thailand’s export represents that Thailand exports products to Japan. ??? P7, L255; This result consists of the work of Tinbergen, 1962; …. What is the meaning? Also, starting sentences with “After that”, “Then” should be avoided P4, L14; Remove external square bracket P6, L198, TFP is not defined P1 and others!! What is Thai export, Thai GDP? Find difference between Thai and Thailand!!

Response 4 We have corrected as you suggested.

Point 5 The results need to be more clarified: The authors should result for the ADF and PP to see how the variables became stationary after first difference. Does regime 1 corresponds to high growth market? Add respectively to end the sentence on P7, L222. Tables 2 and 3: The variables do not correspond to the variables defined for the equations. I suppose The TH and JP as used in the Tables must be subscript. Please check and correct. Also, “For the Thailand’s export equation in the high growth market, the Thai GDP and the level of Thailand’s trade openness affect positively Thai exports”  How? It is not clear what indicators support this assertion.

Response 5 We add the unit root test in table 1. The equation and the TH and JP subscript were rejected and rewrote all the results for both export and import equations.

Point 6 Authors should explain statement with results from the table. The authors should do the following: 1) Elaborate more on the results from the table with economic intuition. Stating regression results only and citing literature works to back result is not enough. 2) Authors should comment on the implication of the coefficient of other variables. Example, what does the negative correlation of DIST and TFP to export in Regime implies? 2). Results in Regime are completely left out in the paper. Can the authors explain what is happening in Regime 2 for readers to know? P8, L247; “Table 4 shows Markov Switching probability in regime 1 and regime 2. Coefficients of Markov Switching Probability in regime 1 and regime 2 are very similar which are 0.323 and 0.323, respectively.”

Response 6 For the empirical result, we add more explanation.

Point 7 How does this result statement add to contribution in the paper? Regimes 1 and 2 have similar Coefficients of Markov Switching Probability means what? The phrase “boost decisively” as used in the abstract and conclusion is misleading. It tends to imply causality which is not backed by the result. The authors should write as obtained from the results … “positive correlation”. 

Response 7 We have corrected. 

Reviewer 2 Report

Thank you for the possibility to read this paper. After reading the article carefully, I have some suggestions for its improvement:

It’s not clear how this paper contributes to the existing literature on bilateral trade? The introduction section should present what has already been done by other researches in this field and indicate the existing gap of knowledge (limitation of empirical researches). What is the novelty and originality of this paper? I would like to suggest including literature review section, as now there is no background for explanatory variables selection, i.e. why explanatory variables are GDP, population, distance, technological progress and trade openness? Why, for example, exchange rate is not included? Formula (1) suggests that imports and exports depend on the GDP of the country of origin and the GDP of the partner country. However, in equation (2) Thailand’s exports depend on Thailand’s GDP but not on Japan’s GDP. Why? The literature on international trade suggests that country’s exports are determined by the size of its trading partner’s economy, thus one may expect that Japan’s GDP and Japan’s population growth can contribute to Thailand’s exports to Japan. Similarly, the market size of Thailand (GDP and population growth) can influence its imports from Japan. The explanation why these variables are omitted should be presented in the paper. In addition, I would suggest including a justification as to why the factors chosen affect imports (exports), for example, why population growth in Japan should have an impact on Thailand’s imports? In Al-Majali and Adayleh (2018) imports and exports of country i depend on trade openness of country i; in this paper Thailand’s imports depend on Japan’s openness. An explanation of this choice would be appropriate. The results indicate, that economic size of Japan (GDP) and Thailand’s import has positive relation as in Al-Majali and Adayleh (2018) (lines 241-242). But in Al-Majali and Adayleh (2018) imports of country i depend on GDP of country i, not on the GDP of trading partner as it is in this paper. The source of TFP data should be more clearly identified, as “from the conference board website” is not very informative. How the reader can find this website? The measurement unit of data used should be presented. One abbreviation of total factor productivity should be used, TFP or TEP. The level of trade openness of Thailand is calculated by formula, presented in line 92, shouldn’t here be total export of Thailand instead of Japan? Also, I suggest numbering all formulas. The estimates are presented for two regimes, however, only results for regime 1 are discussed in sections 4 and 5. In addition, what is the theoretical background to expect different exports and imports reaction to openness in fast and slow growing markets?

Author Response

Response to Reviewer 2 Comments

Point 1: It’s not clear how this paper contributes to the existing literature on bilateral trade? The introduction section should present what has already been done by other researches in this field and indicate the existing gap of knowledge (limitation of empirical researches).

Response 1: There are some studies investigating the relationship between the trade openness and economic growth such as Yanikkaya, H. (2003). However, there is no research investigating the relationship between trade openness and bilateral trade. Thus, this is the first study of this topic.

Reference:

Trade openness and economic growth: a cross-country empirical investigation. Journal of Development economics, 72(1), 57-89.

Point 2: What is the novelty and originality of this paper? I would like to suggest including literature review section, as now there is no background for explanatory variables selection, i.e. why explanatory variables are GDP, population, distance, technological progress and trade openness? Why, for example, exchange rate is not included? 

Response 2: Since we know that trade drives exports. So, international trade policy uses trade openness as a tool to drives exports. However, there is no direct study investing the relationship between trade openness and exports.

We added more background for explanatory variables. Furthermore, we added the exchange rate variable to our model. However, we cut the TFP and population variables out because we can find the exchange rate in quarterly basis. The available data of TFP and population is only in yearly basis. To gain more information from the data, we decide to not include TFP and population in our model.

Point 3: Formula (1) suggests that imports and exports depend on the GDP of the country of origin and the GDP of the partner country. However, in equation (2) Thailand’s exports depend on Thailand’s GDP but not on Japan’s GDP, Why? The literature on international trade suggests that country’s exports are determined by the size of its trading partner’s economy, thus one may expect that Japan’s GDP and Japan’s population growth can contribute to Thailand’s exports to Japan. Similarly, the market size of Thailand (GDP and population growth) can influence its imports from Japan. The explanation why these variables are omitted should be presented in the paper.

Response 3: We have corrected. We have included the GDP of the partner country.

Point 4 In addition, I would suggest including a justification as to why the factors chosen affect imports (exports), for example, why population growth in Japan should have an impact on Thailand’s imports? In Al-Majali and Adayleh (2018) imports and exports of country i depend on trade openness of country i; in this paper Thailand’s imports depend on Japan’s openness. An explanation of this choice would be appropriate. The results indicate, that economic size of Japan (GDP) and Thailand’s import has positive relation as in Al-Majali and Adayleh (2018) (lines 241-242). But in Al-Majali and Adayleh (2018) imports of country i depend on GDP of country i, not on the GDP of trading partner as it is in this paper.

Response 4 We have corrected as you suggested. We have added as following

Applying the gravity model in international trade, the gravity equation consists of the “mass” factor represented by the economic sizes, and the distance referring to distance between two countries (Guo, 2015). Most studies use the GDP as a proxy for demand and supply measuring of a country’s economic and market size (Kepaptsoglou, Konstantinos; Karlaftis, Matthew; Tsamboulas, Dimitrios, 2010). As the domestic industries grow in size and productivity, the demand for the goods produced by them rises in the international market. (Ronit, and Divya, 2014). Some studies (Martínez-Zarzoso, & Nowak, 2003; Simwaka, Kisu, 2006; Achay, 2006; Arabi, & Ibrahim, 2012) pointed out that the GDP is found to be important determinants of bilateral trade flows and have a positive impact on the volume of trade.

Point 5 The source of TFP data should be more clearly identified, as “from the conference board website” is not very informative. How the reader can find this website? The measurement unit of data used should be presented. One abbreviation of total factor productivity should be used, TFP or TEP. The level of trade openness of Thailand is calculated by formula, presented in line 92, shouldn’t here be total export of Thailand instead of Japan? Also, I suggest numbering all formulas.

Response 5 We decides to cut TFP variable off because we would like to obtain quarterly data, and the available data of TFP is only in yearly basis. As we add exchange rate variable, and it provides in quarterly data. Thus, to gain more information, we employ quarterly data.

Point 6 The estimates are presented for two regimes, however, only results for regime 1 are discussed in sections 4 and 5. In addition, what is the theoretical background to expect different exports and imports reaction to openness in fast and slow growing markets? 

Response 6 We have already added more explanation.

Round 2

Reviewer 1 Report

I find this is a potentially interesting topic but enough room to improve the quality. This manuscript looks interesting but need to revise again. I believe English is not the first language of the authors. I would encourage the authors to improve the paper by first asking a native English speaker or professional editing service to assist what is rather diminishing the contribution of the work. It is difficult to understand phrases like

"Trade openness provides the advantage of decreases the bias of anti-export .."? 

"The following sector is the methodology. Data belongs to section three"

and many more. 

Other areas can be improved as follows: 

1. Title of the paper: I don't think the title of the paper is clear enough (not suitable) to represent the subject matters in manuscript.

2.  Introduction: Introduction still doesn‘t look great. I still do not think the manuscript is situated at the centre trade and growth theory. Without literature review section, this is quite important for the paper. Again, it should be revised to include the clear contribution of the manuscript. 

Author Response

Response to Reviewer 1 Comments

Point 1: I find this is a potentially interesting topic but enough room to improve the quality. This manuscript looks interesting but need to revise again. I believe English is not the first language of the authors. I would encourage the authors to improve the paper by first asking a native English speaker or professional editing service to assist what is rather diminishing the contribution of the work. It is difficult to understand phrases like

"Trade openness provides the advantage of decreases the bias of anti-export .."?

"The following sector is the methodology. Data belongs to section three"

and many more.

Response 1: We have asked a native English speaker to correct our grammar, and we have corrected all of them.

Point 2: Title of the paper: I don't think the title of the paper is clear enough (not suitable) to represent the subject matters in manuscript.

Response 2: We have changed the title of this paper to be “The impact of Thailand's openness on bilateral trade between Thailand and Japan: Copula-Based Markov Switching Seemingly Unrelated Regression Model”.

Point 3 Introduction: Introduction still doesn‘t look great. I still do not think the manuscript is situated at the centre trade and growth theory. Without literature review section, this is quite important for the paper. Again, it should be revised to include the clear contribution of the manuscript.

Response 3: I have revised all of the introduction.

Reviewer 2 Report

I have one important note left in the revised version of the article. It is common to measure trade openness as the ratio of exports and imports to a country's GDP. All the indicators mentioned (exports, imports and GDP) are for the same country. For the first time, I see Thailand's trade openness being calculated by summing up Thai imports and JAPAN's exports. I'm thinking this is either a mistake or needs some explanation.
Part of Table 1 did not fit in the page and I could not see it.

Author Response

Point 1: I have one important note left in the revised version of the article. It is common to measure trade openness as the ratio of exports and imports to a country's GDP. All the indicators mentioned (exports, imports and GDP) are for the same country. For the first time, I see Thailand's trade openness being calculated by summing up Thai imports and JAPAN's exports. I'm thinking this is either a mistake or needs some explanation.

Response 1: This is our mistake. We apologized for our mistake. We have corrected it.

Point 2: Part of Table 1 did not fit in the page and I could not see it.

Response 2: We have written in section 4.1. Augmented Dickey-Fuller (ADF) test and Phillips-Perron (PP) test are used to test unit root in the stationary process. The null hypothesis of the ADF and PP test is that the time-series is stationary and has a unit root. Table 1 shows the unit root test results of the ADF and PP test statistics at the level and 1st difference. Wasserstein and Lazar (2016) from the American Statistical Association (ASA) demonstrated that p-value does not provide a good measurement in statistic testing. As a consequence, this study presents the Bayes Factors as an alternative approach for testing accurate null hypotheses. Overall, the Bayes factor provides some evidence to support that all variables are stationary at the first difference.